# Body Fat Changes and Liver Safety in Obese and Overweight Women Supplemented with Conjugated Linoleic Acid: A 12-Week Randomised, Double-Blind, Placebo-Controlled Trial

**DOI:** 10.3390/nu12061811

**Published:** 2020-06-17

**Authors:** Edyta Mądry, Ida Judyta Malesza, Mehala Subramaniapillai, Agata Czochralska-Duszyńska, Marek Walkowiak, Anna Miśkiewicz-Chotnicka, Jarosław Walkowiak, Aleksandra Lisowska

**Affiliations:** 1Department of Physiology, Poznan University of Medical Sciences, 61701 Poznań, Poland; m.subram@gmail.com (M.S.); czochralska@gmail.com (A.C.-D.); 2Department of Pediatric Gastroenterology and Metabolic Diseases, Poznan University of Medical Sciences, 61701 Poznań, Poland; ida.malesza@gmail.com (I.J.M.); chotnicka@ump.edu.pl (A.M.-C.); jarwalk@ump.edu.pl (J.W.); 3Division of Reproduction, Department of Gynecology and Obstetrics, Poznan University of Medical Sciences, 61701 Poznań, Poland; walkowiak.gpsk@gmail.com; 4Department of Clinical Auxology and Pediatric Nursing, Poznan University of Medical Sciences, 61701 Poznań, Poland; alisowska@ump.edu.pl

**Keywords:** liver, cytochrome P450, ^13^C-methacetin breath test, obesity, adipose tissue, body composition

## Abstract

Preliminary evidence suggests that conjugated linoleic acid (CLA) may reduce body weight and affect body composition. The present study assessed the effect of CLA supplementation on body fat composition in overweight and obese women, while also evaluating the liver safety of CLA use. Seventy-four obese or overweight women were randomly assigned to receive 3 g/day CLA or placebo for 12 weeks. Body composition (dual-energy X-ray absorptiometry) and liver function (^13^C-methacetin breath test and serum liver enzymes) were assessed before and after the trial. Patients receiving CLA experienced a significant reduction of total body fat expressed as mass (*p* = 0.0007) and percentage (*p* = 0.0006), android adipose tissue (*p* = 0.0002), gynoid adipose tissue (*p* = 0.0028), and visceral adipose tissue (*p* = 4.2 × 10^−9^) as well as a significant increase in lean body mass to height (*p* = 6.1 × 10^−11^) when compared to those receiving a placebo. The maximum momentary 13C recovery changes and end-point values were significantly higher in the CLA group when compared to the placebo group (*p* = 0.0385 and *p* = 0.0076, respectively). There were no significant changes in alanine aminotransferase, asparagine aminotransferase, and gamma-glutamyl transpeptidase activities between the groups. In conclusion, CLA supplementation was well tolerated and safe for the liver, which shows beneficial effects on fat composition in overweight and obese women.

## 1. Introduction

Overweight and obesity conditions are increasing in prevalence in both developing and developed countries [1], which is a cause for concern as excess adiposity is associated with many different long-term health complications, including cardiovascular disease, diabetes, and various forms of cancer [2] In particular, visceral fat has been correlated with worse health outcomes and increased mortality risk, particularly among women [3]. Therefore, the identification of strategies that may help reduce adiposity, particularly visceral adiposity, has the potential to curb these adverse health outcomes and promote well-being.

Diet supplementation with conjugated linoleic acid (CLA) may be an interventional strategy to reduce adiposity among obese and overweight individuals. CLA refers to a group of geometric and positional isomers of linoleic acid with a conjugated double bond. The highest concentrations of CLA are found in the meat of ruminants and dairy products. Although humans can endogenously produce CLA, the concentrations in blood and tissues of non-supplemented individuals are low [4,5]. The potential benefits of CLA supplementation include anti-cancerogenic activity [6], fat and body weight reduction, inhibition of atherogenesis, and anti-diabetic effects [7].

Previous studies have evaluated the use of CLA for body fat reduction [8,9,10,11]. However, various methodological shortcomings have limited interpretability of the study findings. Blankson et al. [8] conducted a randomised, double-blind, placebo-controlled trial evaluating the effect of various doses of CLA (1.7, 3.4, 5.1, or 6.8 g CLA/d) compared to 9 g of olive oil among 60 overweight and obese adults. The authors reported a statically significant reduction in body fat mass in the groups receiving 3.4 g and 6.8 g of CLA when compared to an alternative intervention. However, these results should be interpreted with caution due to the small sample size within each group (*n* =12). Gaullier et al. [9] also conducted a double-blind, placebo-controlled trial of 118 participants to study the effects of CLA supplementation on fat mass and its distribution compared to the placebo for six months, finding a significant decrease in body fat mass at 3 and 6 months. The changes were visible, particularly in women, and were more pronounced in the lower extremities of the participants. In 2012, Carvalho et al. [10] tested a similar hypothesis among 14 women diagnosed with metabolic syndrome, where the participants randomly received micro-encapsulated CLA (3 g/day) or a placebo. There were no significant between-group differences at any point during the study. In the CLA group, there was a significant reduction in percent body fat mass after 30, 60, and 90 days of supplementation with a statistically significant decrease in the placebo group after 90 days. The intervention was associated with a hypocaloric diet. Therefore, the changes observed might also be attributable to reduced caloric intake as opposed to exclusive CLA supplementation. Another major limitation of this study was the small sample size.

In addition to efficacy, it is important to consider and evaluate the liver safety profile of CLA supplementation. One of the diseases related to obesity is non-alcoholic fatty liver disease (NAFLD) [12], which affects about 90% of patients with severe obesity undergoing bariatric surgery [13]. According to the available literature, up to 5% of obese patients may have unsuspected cirrhosis [14] and it has been suggested that NAFLD ranges from 57% of overweight out-patient clinics visitors to 98% of nondiabetic obese individuals [15,16,17]. There is some evidence from animal studies to suggest that CLA supplementation may inhibit CYP1A2 activity, which is a key member of the cytochrome P40 family of enzymes in the liver involved in drug metabolism [18].

This data analysis aimed to explore the effect of 12-week supplementation of CLA in overweight and obese women on fat reduction, particularly visceral fat, and to assess efficacy and safety of this supplementation on liver function.

## 2. Methods

### 2.1. Patient Characteristics

Obese or overweight female participants were recruited between July 2014 and May 2015 via the Departments of Poznan University of Medical Sciences, Poland. Inclusion criteria were as follows: women over the age of 18 years and body mass index (BMI) ≥25 kg/m^2^. Exclusion criteria comprised a history of chronic systemic disease (with the exception of hypertension), celiac disease, type 2 diabetes, pancreatic and/or liver disease, current or recent (within the preceding month) treatment with CLA, or agents interfering with fat digestion and/or absorption (chitosan, orlistat, green tea) and pregnancy [19].

Of the 187 volunteers, 81 met the inclusion criteria, with seven participants excluded due to abdominal pain and diarrhoea (1), difficulties in cooperation (1), personal problems (1), shortage of time (3), and suspicion of ovarian tumour (1). Seventy-four women underwent randomisation and were included in the current data analyses. The dropout percentages in the CLA and placebo groups were identical (18.9%) and comprised participants who: (1) did not appear for the final study visit (3 in CLA group vs. 4 in the placebo group), (2) reported adverse effects of the tested product (nausea: 2 in the placebo group, rash: 1 participant from the placebo group), (3) became pregnant (2 participants from the CLA group), and (4) did not arrive for the breath test (2 participants from the CLA group).

At baseline, all participants underwent a physical examination, including the evaluation of body weight, height, and body mass index (BMI). Baseline characteristics of the study group are presented in Table 1 with no between-group differences in any of the baseline characteristics.

### 2.2. Study Design

The protocol of this randomised, double-blind, placebo-controlled nutritional intervention was previously described by Lochocka et al. [19]. The study participants were randomly assigned to receive CLA or placebo. CLA and placebo products (Olimp Laboratories, Debica, Poland) were identical, transparent capsules containing 0.5 g of 80% CLA (50:50 cis-9, trans-11, and trans-10, cis 12 isomers) or 0.5 g of sunflower oil (placebo). The fatty acid composition of both capsules have been detailed in our previous paper [20].

During randomisation, a computer-generated number was assigned to each participant with study participants and investigators blinded to the intervention assignment. All the study participants were instructed to take two capsules of the given product three times per day (3 g of CLA or placebo) for 12 weeks. The completion criterion was the consumption of 75% of the capsules provided. Participants were asked to maintain their lifestyle for the duration of the study, including physical activity and eating patterns. Before and after the completion of the study, participants underwent a full-body scan using a dual-energy X-ray absorptiometry (densitometry), venous blood collection, and the ^13^C-methacethin breath test (^13^C-MBT).

Total body fat, android (fat distribution along the trunk/upper body), gynoid (fat distribution around the hips, breasts, and thighs), visceral adiposity, and lean body mass were measured using a densitometric full-body scan (Hologic Discovery, Marlborough, MA, USA). A venous blood sample was collected according to standard methods following an overnight fast. The serum was stored at −70 °C until analysis. Activities of alanine aminotransferase (ALT), asparagine aminotransferase (AST), and gamma-glutamyl transpeptidase (GGT) were determined by the kinetic method at 37 °C (AU480 Chemistry Analyser, Beckman Coulter, Brea, CA, USA).

The ^13^C-MBT breath test was applied to assess changes in liver function by monitoring cytochrome P450 isoenzyme CYP1A2 activity. This test was conducted at rest after an overnight fast. The participants were instructed to avoid 13C-naturally rich products such as kiwi, cane sugar, pineapple, and maize for 48 h preceding the test [21]. After basal breath sample collection (designated as “0”), every subject received 75 mg of ^13^C-methacetin (Campro Scientific GmbH, Berlin, Germany) dissolved in 200 mL of unsweetened fruit tea. During two hours of the test, 10 samples of expiratory air were obtained (respectively: 0 [basal], 10, 20, 30, 40, 50, 60, 80, 100, and 120 min after substrate intake). The breath samples were collected in plastic bags, closed immediately after exhalation, and stored for the analysis. The ^13^CO_2_ concentrations in the breath samples were measured using isotope-selective nondispersive infrared spectrometry (IRIS, Wagner Analysen Technik GmbH, Bremen, Germany) [21]. From the obtained curve, the following parameters were derived: the maximum momentary ^13^C recovery (Dmax), the time of maximum Dmax occurrence (Tmax), and the cumulative percentage dose recovery at 120 min of the test (CPDR).

The project was supported by NUTRICIA Foundation’s research grant (number: 504-05-01103115-000-15-07588) and the grant from Poznan University of Medical Sciences, Poland (502-0101103115-07588). The design, analysis of study results, and the writing of this article were not affected by the NUTRICIA Foundation. Research was conducted according to the guidelines from the Declaration of Helsinki. Informed written consent was obtained from every study participant and the local Bioethics Committee of the Institutional Review Board at Poznan University of Medical Sciences, Poland (approval number 606/12) accepted the study protocol. The trial was registered in the German Clinical Trials Register (DRKS-ID: DRKS00010462).

### 2.3. Statistical Analysis

The results are presented as means and standard deviations (SD) and medians with interquartile ranges. Between-group differences in baseline parameters were calculated using the Mann Whitney U test. Changes (baseline vs. 12 weeks) in the laboratory and 13C-MBT parameters observed in the groups (CLA vs. placebo) were compared using the Mann Whitney U test with the Wilcoxon-rank test used for within-group analyses (baseline vs. 12 weeks). The significance level was set at *p* < 0.05. Differences between two groups of categorical variables were assessed with two-tailed Fisher’s exact test. Statistical analysis was performed with STATISTICA 12 software packages (StatSoft Inc., Tulsa, OK, USA). Assuming a potential dropout rate of 20%, a significance level of 5% and the detection power of 80%, the sample size was calculated as 37 for each group [19,20].

## 3. Results

At baseline, there were no between-group differences in the assessed parameters (Table 2).

### 3.1. Fat and Lean Mass Composition

Changes in body fat composition and lean body mass are compared in Table 3. Patients who received CLA for 12 weeks had a significant reduction in multiple parameters of adiposity compared to women who received a placebo including total body fat expressed as mass (*p* = 0.0007) and percentage (*p* = 0.0006), android adipose tissue (*p* = 0.0002), gynoid adipose tissue (*p* = 0.0028), and visceral adipose tissue (*p* = 4.2 × 10^−9^). There was a significant increase in lean body mass to height in the CLA group when compared to the placebo group (*p* = 6.1 × 10^−11^).

Significantly more participants (*p* = 0.0038) in the CLA group (*n* =27, 84.4%) had a reduction in total body fat expressed as mass compared to those in the placebo group (*n* =15, 50.0%). Similarly, more participants (*p* = 0.000058) in the CLA group (*n* =31, 96.9%) had a decrease in fat percentage when compared to subjects in the placebo group (*n* =16, 53.3%) at the end of the trial. Twenty-nine (90.6%) participants in the CLA group and 14 (46.7%) subjects in the placebo group had a reduction in visceral adipose tissue (*p* = 0.00024). Similarly, the intervention resulted in an increase in lean body mass in 31 women (96.9%) in the CLA group and 13 (43.3%) in the placebo group (*p* = 0.000058).

### 3.2. Liver Function

The changes (Table 4) and end-point values of Dmax (Figure 1) after 12 weeks of nutritional intervention were significantly higher in the CLA group when compared to the placebo group (*p* = 0.0385 and *p* = 0.0076, respectively). The remaining parameters of the breath test were not different between the groups studied. Similarly, changes of the liver enzyme activities (ΔAST, ΔALT, ΔGGTP) did not differ between the placebo and CLA groups (Table 4).

## 4. Discussion

The results of this double-blind, randomised, placebo-controlled trial demonstrated that overweight and obese women taking CLA over a period of 12 weeks had reductions in multiple fat parameters including visceral, android, and gynoid adipose tissues. This is the first robust study, to our knowledge, to demonstrate a significant decrease in adipose parameters, which concurrently demonstrates liver safety. Therefore, the use of CLA supplementation adjunctively to a healthy, well-proportioned diet may support weight loss efforts among overweight and obese individuals.

With regard to liver function, the gain in Dmax parameters indicates an increased CYP1A2 enzyme activity, which is contrary to the hypothesis that CYP1A2 is inhibited by CLA as reported by Josyula et al. in rats [18]. The CLA supplementation in animal models results in liver enlargement and hepatic steatosis [22,23,24,25,26]. The negative effect on the liver of mice and hamsters particularly resulted from trans10, cis12 CLA supplementation [24,25,26,27]. Although the results of animal studies indicate the adverse effect of CLA on liver function, a systematic review and meta-analysis by Mirzaii et al. (2016) reported inconclusive results [28]. Thirteen RCTs examined the effect of CLA consumption (either CLA supplements or CLA-enriched foods) on liver function (i.e., serum AST, ALT, and ALP activities) among healthy participants and did not find any significant relationship between CLA consumption and ALT or ALP levels [28]. However, CLA supplementation significantly increased AST activities in comparison to the placebo group (mean difference = 0.171 U/L, 95% CI, 0.034–0.307, *p* = 0.01). Furthermore, to date, there are only three separate case reports of acute liver failure or toxic hepatitis due to CLA supplementation [29,30,31]. However, the reports mentioned above were unrelated to each other, and the exact dose and composition of ingested CLA are unknown. Moreover, the relationship might be coincidental. The discrepancies between animal and human studies may be associated with differences in the mechanisms responsible for liver damage between species. Furthermore, in most human studies, an equal mixture of the cis9, trans11, and trans10, cis12 CLA isomers was used, so it is possible that the predominance of the cis9, trans11 CLA isomer in humans may overshadow the adverse effect of trans10, cis12 CLA isomer observed in animal models [7,29].

Multiple mechanisms have been proposed to explain the effects of CLA on metabolism and body composition. However, it is not fully established yet. Data come from research on animal models and cannot be simply extrapolated on humans. CLA supplementation contributes to an increase of lean body mass [32] and fat-mass loss likely by lowering the activities of lipoprotein lipase and Δ9-desaturase, and, thereby, reducing lipid uptake into adipocytes rather than enhancing lipolysis [33,34,35]. Some research studies also suggested that CLA affects preadipocyte differentiation [36] and can stimulate adipocyte’s apoptosis [37]. Moreover, CLA also has a stimulative impact on energy expenditure, possibly by the upregulation of uncoupling proteins (UCPs) expressed in mitochondria of various tissues such as the adipose, liver, and the skeletal muscle [38]. Close et al. reported that patients who received supplements with 4 g of CLA had significantly increased fat oxidation and energy expenditure during sleep [39]. On the other hand, there is evidence that CLA supplementation may promote fatty liver as a consequence of reduced glucose disposal mediated by adipose tissue and enhanced transport of fatty acids to the hepatocytes in mice [40,41]. However, there seem to be significant differences between species, as supplementation of CLA in hamsters lead to liver hypertrophy but not lipid accumulation. In addition, rats and pigs fed after CLA supplementation showed no changes in weight or lipid content in the liver [42]. There is still a lack of conclusive evidence whether the CLA-enriched diet can exhibit adverse effects on human liver.

To the authors’ best knowledge, this is the first study assessing the efficacy, tolerability, and safety of CLA supplementation on liver function simultaneously using serum markers and the ^13^C-MBT breath test. A key strength of this study is its robust study design in evaluating the benefits of CLA supplementation in women using a double-blind, placebo-controlled trial. Regarding the sex differences, it is also important to design and evaluate these study parameters in men. A limitation of this study was that blood CLA levels were not assessed. The intervention period (12 weeks) was considered by authors to be sufficient to observe body composition changes and possible short-term changes in liver function. However, for evaluating long-term hepatic influence, it could be insufficient. Liver disease was an exclusion criterion in the present study. Patients with hepatic involvement could have abnormal results of the methacetin breath test and abnormal activities of liver enzymes. The inference from the study comprising such patients could be limited due to the potential masking effect of initial liver status. On the other hand, knowing that short-term CLA supplementation is safe for the liver, one could plan the evaluation of the long-term CLA effect and the impact on initially abnormal liver function. Diabetes has been ruled out as a disease severely disturbing the energy balance by affecting carbohydrates and fat metabolism. However, since a significant percentage of overweight and obese subjects present glucose metabolism disturbances, the conclusions for the general population are limited.

In conclusion, CLA supplementation was well tolerated and safe for the liver while, at the same time, showing beneficial effects on fat composition in overweight and obese women.

## Figures and Tables

**Figure 1 nutrients-12-01811-f001:**
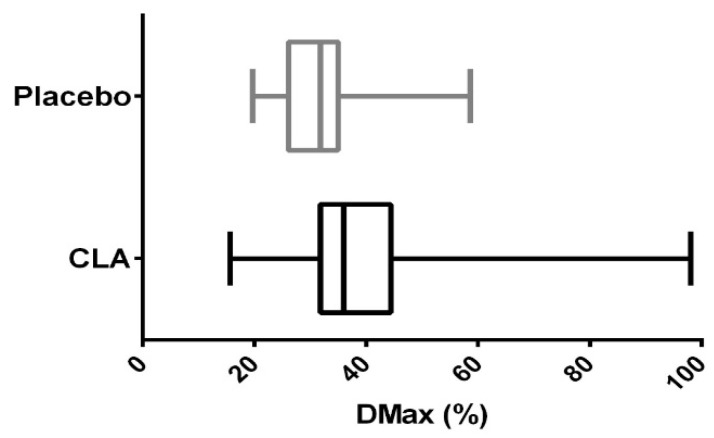
Maximum momentary ^13^C recovery (Dmax) in subjects receiving conjugated linoleic acid (CLA) and placebo after 12 weeks of nutritional intervention (*p* = 0.0076).

**Table 1 nutrients-12-01811-t001:** Baseline characteristics of the study groups.

	CLA Group (*n* =32)		Placebo Group (*n* =30)		*p*
	Median	1st–3rd Quartile	Median	1st–3rd Quartile	
Age (years)	54	43–59	54	45–61	0.9914
Body weight (kg)	90.3	77.6–97.7	93.5	81.4–100.7	0.3967
Body height (cm)	163	160–166	163	159–167	0.7068
BMI (kg/m^2^)	34.00	30.70–37.58	35.36	31.75–38.62	0.5049

CLA–conjugated linoleic acid, *p*–statistical significance, BMI–body mass index.

**Table 2 nutrients-12-01811-t002:** Baseline characteristics of the studied parameters.

	CLA Group (*n* =32) *		Placebo Group (*n* =30)		*p*
	Median	1st–3rd Quartile	Median	1st–3rd Quartile	
Total body fat (kg)	41.35	34.38–47.16	42.21	34.33–48.62	0.8833
Total body fat (%)	44.6	42.5–48.5	44.1	41.4–47.7	0.6696
Android adipose tissue (kg)	3.728	3.008–4.233	3.782	2.945–4.307	0.7958
Gynoid adipose tissue (kg)	6.103	4.978–7.151	6.527	5.372–7.259	0.5151
Visceral adipose tissue (g)	976	819–1278	1024	837–1144	0.8833
Lean body mass/height^2^ (kg/m^2^)	17.5	16.4–18.5	18.1	16.6–18.9	0.4126
Appendicular lean body mass/height^2^ (kg/m^2^)	7.80	7.23–8.30	7.87	7.25–8.49	0.8722
Dmax (%)	35.9	29.2–42.0	34.6	28.1–39.9	0.4853
Tmax (min)	20.0	20.0–20.0	20.0	10.0–20.0	0.1772
CPDR (% ^13^C)	32.1	29.9–38.6	33.7	29.8–36.1	0.9941
ALT (IU/l)	23.0	19.1–26.2	23.1	20.3–25.5	0.9707
AST (IU/l)	22.5	17.3–29.8	21.5	19.0–28.8	0.6757
GGTP (IU/l)	22.9	20.0–31.1	22.6	16.8–32.3	0.5133

* *n* =30 for ^13^C-MBT parameters (Dmax, Tmax, CPDR) CLA–conjugated linoleic acid, *p*–statistical significance, Dmax–the maximum momentary ^13^C recovery, Tmax–time to reach Dmax, CPDR–the cumulative percentage dose recovery at 120 min. ALT–alanine aminotransferase, AST–asparagine aminotransferase, GGT–gamma-glutamyl transpeptidase.

**Table 3 nutrients-12-01811-t003:** Effects of conjugated linoleic acid (CLA) and placebo on body composition outcome measures.

	CLA Group (*n* =32)		Placebo Group (*n* =30)		*p*
	Median	1st–3rd Quartile	Median	1st–3rd Quartile	
**∆**Total body fat (kg)	−1.58	−2.24–−0.72	0.07	−0.96–1.69	0.0007
**∆**Total body fat (%)	−1.55	−2.30–−1.20	−0.20	−1.65–1.03	0.0006
**∆**Android adipose tissue (kg)	−0.31	−0.46–−0.19	−0.09	−0.21–0.05	0.0002
**∆**Gynoid adipose tissue (kg)	−0.21	−0.43–−0.07	0.11	−0.22–0.33	0.0288
**∆**Visceral adipose tissue (kg)	−0.11	−0.16–−0.07	0.01	−32.5–50.25	4.2 × 10^−9^
**∆**Lean body mass/height (kg/m^2^)	1.3	0.9–1.625	−0.1	−0.3–0.275	6.1 × 10^−^^11^
**∆**Appendicular lean body mass/height^2^ (kg/m^2^)	0.2	−0.1175–0.4475	0.035	−0.3025–0.335	0.1201

*p*–statistical significance.

**Table 4 nutrients-12-01811-t004:** Effects of conjugated linoleic acid (CLA) and placebo on liver outcome measures.

	CLA (*n* =32) *		Placebo (*n* =30)		*p*-Value
	Median	1st–3rd Quartile	Median	1st–3rd Quartile	
ΔDmax	4.7	−2.8–9.2	−2.1	−10.4–5.0	0.0385
ΔTmax	0	0–0	0	0–10	0.2601
ΔCPDR	−1.2	−2.9–3.2	−0.6	−4.0–2.8	0.5325
ΔALT	1.0	−1.0–3.0	1.0	−1.8–3.0	0.8891
ΔAST	0.5	−4.3–5.3	1.0	−3.0–5.8	0.8315
ΔGGTP	0.0	−3.8–3.0	0.5	−1.0–3.5	0.7082

* *n* = 30 for ^13^C-MBT parameters (Dmax, Tmax, CPDR), Dmax–the maximum momentary ^13^C recovery, Tmax–time to reach Dmax, CPDR–the cumulative percentage dose recovery at 120 min, CLA–conjugated linoleic acid, *p*–statistical significance. ALT–alanine aminotransferase, AST–asparagine aminotransferase, GGT–gamma-glutamyl transpeptidase.

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
