# Peer review of "Body Fat Changes and Liver Safety in Obese and Overweight Women Supplemented with Conjugated Linoleic Acid: A 12-Week Randomised, Double-Blind, Placebo-Controlled Trial"

_nutrients, 2020, doi:10.3390/nu12061811_

Round 1
Reviewer 1 Report
The manuscript by Mądry et al, entitled “Body fat changes and liver safety in obese and overweight women supplemented with conjugated linoleic acid: a twelve-week randomised, double-blind, placebo-controlled trial” is of simple experimental design and may be scientifically interesting. However, there are major concerns that severely affect the work.
- Overall, the manuscript is written in an approximate way, not giving the right value to the data presented.
- The blood CLA levels were not assessed and this is a great limitation of the study. The Authors already know this problem.
- The section “Patient characteristics” is not detailed, could be improved.
- When you talk about “Total body fat, android (fat distribution along the trunk/upper body), gynoid (fat distribution around the hips, breasts and thighs), visceral adiposity and lean body mass were measured.” You should explain how you did these measurements.
- Why you decide to enroll only women in the study? There is not an explanation of this decision. In the paper that you cited (Lochocka et al., 2014) you talked about “adult volunteers” and in this study you have enrolled only female volunteers. Male-female differences should be carefully evaluated.
Minor points
- In the Abstract section there are acronyms, without explanation (e.g. AST, ALT, and GGTP). In the manuscript the same for other acronyms (e.g. 13C-MBT).
- Table 2 and Table 3 are not clear, don’t use brackets.
Author Response
We would like to thank the Reviewer for their thoughtful comments.
In terms of English editing, the first version of the submitted manuscript had been revised by the proper proof-reading service.
Reviewer 1
Comments and Suggestions for Authors
The manuscript by Mądry et al, entitled “Body fat changes and liver safety in obese and overweight women supplemented with conjugated linoleic acid: a twelve-week randomised, double-blind, placebo-controlled trial” is of simple experimental design and may be scientifically interesting. However, there are major concerns that severely affect the work.
- Overall, the manuscript is written in an approximate way, not giving the right value to the data presented.
The data had been presented as baseline characteristics and changes (D), which are typical endpoints for RCT. We changed the manuscript according to all suggestions. Among the other we changed the baseline charcteristics from „intention-to-treat” to „per protocol” mode.
- The blood CLA levels were not assessed and this is a great limitation of the study. The Authors already know this problem.
As we mentioned in the section limitations of the study, blood CLA levels were not assessed. Unfortunately, we can not evaluate this parameter since we do not have the respective samples anymore.
- The section “Patient characteristics” is not detailed, could be improved.
We divided Table 1 and we moved its first to the section Patient characteristics. (The second part was left in the section Results. This action has respectively changed the numbering of subseqent tables).
- When you talk about “Total body fat, android (fat distribution along the trunk/upper body), gynoid (fat distribution around the hips, breasts and thighs), visceral adiposity and lean body mass were measured.” You should explain how you did these measurements.
Body composition was assessed using dual-energy X-ray absorptiometry. All participants underwent a full-body scan using a densitometer (Hologic Discovery, Marlborough, Massachusetts, United States). This information was included in the paragraph describing the design of the study. We added this information to the section describing body composition assessment.
- Why you decide to enroll only women in the study? There is not an explanation of this decision. In the paper that you cited (Lochocka et al., 2014) you talked about “adult volunteers” and in this study you have enrolled only female volunteers. Male-female differences should be carefully evaluated.
Initially, we considered studying subjects with no regard for sex. However, after the preliminary search for volunteers, it appeared that almost exclusively women were willing to participate in the study. Therefore, we considered to take it as an advantage and limited the study to women.
We fully agree with Reviewer 2 that “male-female differences should be carefully evaluated.” We referred to that in the originally submitted manuscript in the previous the last paragraph of the Discussion: „Regarding the sex differences, it is also important to design and evaluate these study parameters in men.”
Minor points
- In the Abstract section there are acronyms, without explanation (e.g. AST, ALT, and GGTP). In the manuscript the same for other acronyms (e.g. 13C-MBT).
The proper changes have been introduced.
- Table 2 and Table 3 are not clear, don’t use brackets.
The brackets have been removed.
Reviewer 2 Report
In the article “Body fat changes and liver safety in obese and overweight women supplemented with conjugated linoleic acid: a twelve-week randomised, double-blind, placebo-controlled trial” the authors analyzed the effects of conjugated linoleic acid (CLA) on parameters of body composition. Since there are reports of negative effects of CLA on the liver, the authors also tested safety of this intervention by biomarkers of hepatic health. They found a reduction of body fat and an increase of lean body mass. While liver enzymes were not elevated the activity of CYP1A2 were even increased which was contrary to the expected inhibition of this enzymes.
The paper was clearly written and understandable and the topic is worth investigating. However, some minor issues have to be addressed before it is ready for publication.
First, the baseline characteristics should be presented for the participants that were also included in the analysis of the intervention effects since the authors performed a per protocol analysis and not an intention to treat. Therefore, the drop-outs could have changed the comparability of the two groups.
In table 2 the values for the visceral adipose tissue of the placebo group seem to be wrong (far to large).
The limitation section has to be extended. The relatively short intervention period and the exclusion of risk groups (diabetes and underlying liver disease) has to be discussed.
The authors should discuss the potential physiological pathway that could explain the beneficial effects of CLA.
Author Response
We would like to thank the Reviewer for their thoughtful comments.
In terms of English editing, the first version of the submitted manuscript had been revised by the proper proof-reading service.
Reviewer 2
Comments and Suggestions for Authors
In the article “Body fat changes and liver safety in obese and overweight women supplemented with conjugated linoleic acid: a twelve-week randomised, double-blind, placebo-controlled trial” the authors analyzed the effects of conjugated linoleic acid (CLA) on parameters of body composition. Since there are reports of negative effects of CLA on the liver, the authors also tested safety of this intervention by biomarkers of hepatic health. They found a reduction of body fat and an increase of lean body mass. While liver enzymes were not elevated the activity of CYP1A2 were even increased which was contrary to the expected inhibition of this enzymes.
The paper was clearly written and understandable and the topic is worth investigating. However, some minor issues have to be addressed before it is ready for publication.
First, the baseline characteristics should be presented for the participants that were also included in the analysis of the intervention effects since the authors performed a per protocol analysis and not an intention to treat. Therefore, the drop-outs could have changed the comparability of the two groups.
Table 1 has been changed. It now contains the data of patients who were supplemented with either CLA or placebo and finalized the study.
In table 2 the values for the visceral adipose tissue of the placebo group seem to be wrong (far to large).
We would like to apologize for the mistake. We corrected the IQR values.
The limitation section has to be extended. The relatively short intervention period and the exclusion of risk groups (diabetes and underlying liver disease) has to be discussed.
The intervention period (12 weeks) was considered by authors to be sufficient to observe body composition changes and possible short-term changes in liver function. However, we agree that in terms of long-term influence, it could be insufficient.
The primary outcomes of the trial included the evaluation of the effect of CLA supplementation on liver function. Patients with liver diseases could have abnormal results of the methacetin breath test and abnormal activities of liver enzymes. And the inference from the study comprising such patients could be limited due to the potential masking effect of initial liver status. On the other hand, knowing that short-term CLA supplementation is safe for the liver, one could plan the evaluation of the long-term CLA effect and the impact on initially abnormal liver function.
Diabetes has been ruled out as a disease severely disturbing energy balance by affecting carbohydrates and fat metabolism.
The proper information has been added to the limitations of the study.
The authors should discuss the potential physiological pathway that could explain the beneficial effects of CLA.
The proper information has been added to the Discussion:
Multiple mechanisms have been proposed to explain the effects of CLA on metabolism and body composition. However, it is not fully established yet. What is more, much data come from researches on animal models and cannot be simply extrapolated on humans.
CLA supplementation contributes to an increase of lean body mass [1] and fat-mass loss probably by lowering the activities of lipoprotein lipase and Δ9-desaturase, thereby reducing lipid uptake into adipocytes, rather than enhancing lipolysis [2-4]. Some researches also suggested that CLA also affects preadipocyte differentiation [5] and can stimulate adipocyte's apoptosis [6]. Moreover, CLA also has a stimulative impact on energy expenditure, possibly by the upregulation of uncoupling proteins (UCPs) expressed in mitochondria of various tissues, such as the adipose, liver, and the skeletal muscle [7]. Intriguingly, Close et al. reported that patients who received supplements with 4 g of CLA had significantly increased fat oxidation and energy expenditure during sleep [8]. On the other hand, there is evidence that CLA supplementation may promote fatty liver as a consequence of reduced glucose disposal mediated by adipose-tissue and enhanced transport of fatty acids to the hepatocytes in mice [9,10]. However, there seem to be significant differences between species, as supplementation of CLA in hamsters lead to liver hypertrophy but not lipid accumulation, what are more rats and pigs fed after CLA supplementation showed no changes in weight or lipid content in the liver [11]. There is still a lack of conclusive evidence, whether the CLA-enriched diet can exhibit adverse effects on humans liver.
[1] Park Y, Albright K, Liu W, Storkson J, Cook M, Pariza M (1997) Effect of conjugated linoleic acid on body composition in mice. Lipids 32(8):853–858
[2]Pariza MW, Park Y, Cook ME. The biologically active isomers of conjugated linoleic acid. Prog Lipid Res. 2001;40(4):283–98.
[3] Park Y, Albright KJ, Storkson JM, Liu W, Cook ME, Pariza MW. Changes in body composition in mice during feeding and withdrawal of conjugated linoleic acid. Lipids. 1999;34(3):243–8
[4] Choi Y, Kim YC, Han YB, Park Y, Pariza MW, Ntambi JM. The trans-10, cis-12 isomer of conjugated linoleic acid downregulates stearoyl-CoA desaturase 1 gene expression in 3 T3-L1 adipocytes. J Nutr. 2000;130(8):1920–4.
[5] Kang KH, Liu W, Albright KJ, Park Y, Pariza MW. trans-10,cis-12 CLA inhibits differentiation of 3T3–L1 adipocytes and decreases PPAR gamma expression. Biochem Biophys Res Comm. 2003;303:795–9.
[6] Kennedy A, Martinez K, Schmidt S, Mandrup S, LaPoint K, McIntosh M. Antiobesity mechanisms of action of conjugated linoleic acid. J Nutr Biochem. 2010;21(3):171–179.
[7] Ryder JW, Portocarrero CP, Song XM, Cui L, Yu M, Combatsiaris T, Galuska D, Bauman DE, Barbano DM, Charron MJ, Zierath JR, Houseknecht KL. Isomer-specific antidiabetic properties of conjugated linoleic acid—improved glucose tolerance, skeletal muscle insulin action, and UCP-2 gene expression. Diabetes. 2001;50(5):1149–1157.
[8] Close RN, Schoeller DA, Watras AC, Nora EH. Conjugated linoleic acid supplementation alters the 6-mo change in fat oxidation during sleep. Am J Clin Nutr. 2006;86(3):797–804.
[9] Vyas D, Kadegowda AK, Erdman RA. Dietary conjugated linoleic acid and hepatic steatosis: species-specific effects on liver and adipose lipid metabolism and gene expression. J Nutr Metab 2012;2012:932928
[10] Clement L, Poirier H, Niot I, Bocher V, Guerre-Millo M, Krief S, et al. Dietary trans10,cis-12 conjugated linoleic acid induces hyperinsulinemia and fatty liver in the mouse. J Lipid Res 2002;43:1400–9.
[11] Silveira MB, Carraro R, Monereo S, Tébar J. Conjugated linoleic acid (CLA) and obesity. Public Health Nutr. 2007;10(10A):1181‐1186. doi:10.1017/S1368980007000687.
Round 2
Reviewer 1 Report
The paper has been improved according to the comments.